# Geometry aware convolutional filters for omnidirectional images representation

## Abstract

Due to their wide field of view, omnidirectional cameras are frequently used by autonomous vehicles, drones and robots for navigation and other computer vision tasks. The images captured by such cameras, are often analyzed and classified with techniques designed for planar images that unfortunately fail to properly handle the native geometry of such images. That results in suboptimal performance, and lack of truly meaningful visual features. In this paper we aim at improving popular deep convolutional neural networks so that they can properly take into account the specific properties of omnidirectional data. In particular we propose an algorithm that adapts convolutional layers, which often serve as a core building block of a CNN, to the properties of omnidirectional images. Thus, our filters have a shape and size that adapts with the location on the omnidirectional image. We show that our method is not limited to spherical surfaces and is able to incorporate the knowledge about any kind of omnidirectional geometry inside the deep learning network. As depicted by our experiments, our method outperforms the existing deep neural network techniques for omnidirectional image classification and compression tasks.

## 1 Introduction

Drone vision, autonomous cars and robot navigation systems often use omnidirectional cameras, as they allow recording the scene with wide field of view. Despite their obvious advantages, images obtained by such cameras have different statistics compared to planar images. Nevertheless, omnidirectional images are often processed with standard techniques, which are unfortunately poorly adapted to the specific geometry of such images.

In this paper we improve one of the most popular frameworks for image processing, namely convolutional neural network (CNN) for omnidirectional images. CNNs prove to be effective, as they permit to achieve very good performance in many different tasks like image classification, segmentation, generation and compression. In the context of omnidirectional cameras, CNNs are typically applied directly to the unwrapped and distorted spherical images. This approach, however, is suboptimal: due to specific geometry of these images, and, in particular, the change in the image statistics with the position in the image. The latter forces the network to learn different filters for different locations in the omnidirectional images (see Fig. 1).

To solve this issue, we replace ordinary convolutional filters with the graph-based ones that can adapt their size and shape depending on the position in the omnidirectional image thanks to the flexible graph structure. In order to overcome the common limitation of these graph-based filters being isotropic (i.e. invariant to the changes in the objects orientation in the image) we suggest to use multiple directed graphs instead of a single undirected one, as used in (Defferrard et al., 2016; Kipf & Welling, 2017; Khasanova & Frossard, 2017a). This permits our method to encode the orientation of the objects that appear in the images and, therefore, extract more meaningful features from them. This together with the inherent ability of our filters to adapt to the image projective geometry allows our method to reach state-of-the-art performance when working not just with spherical projections, as done by Khasanova & Frossard (2017b); Cohen et al. (2018), but virtually with any type of image projective geometry that can be encoded by a graph. In our experiments we show that apart from the state-of-the-art results on the regular omnidirectional images classification task, our approach outperforms the existing classification techniques when applied to the images projected on a randomly

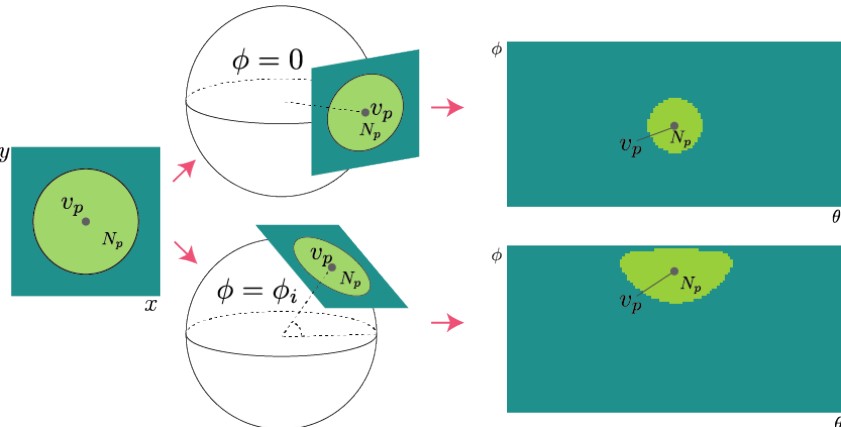

Figure 1: Our network allows adapting the size and shape of the filter with respect to the elevation.

perturbed spherical surface or on a cube via the cube-map projection, which has recently become one of the popular ways to represent 360-images (Chen et al., 2018). Finally, we demonstrate that our method can be applied for orientation-dependent task such as compression and reduce artifacts comparing with a standard approaches.

## 2 RELATED WORK

In this section we first briefly introduce the most recent approaches that combine the power of deep learning methods with graph signal processing techniques, as they are similar in spirit to our work. We then discuss in more details the recent trends in omnidirectional image processing.

**Geometric Deep learning.** In the recent years a number of deep learning methods have been introduced tailored for processing irregular data that is represented as a graph. One way to solve this problem is suggested by Monti et al. (2017), where the authors define a local system of d-dimensional pseudo-coordinates for every node of the graph and learn both the filters and patch operators that work on these coordinates. A different direction is taken by Wang et al. (2018), who propose using edge-based convolutional kernels and dynamically update the graph. While being flexible and effective for general tasks, in the specific context of the omnidirectional images these methods do not directly take the advantage of the knowledge about the projective geometry, which we model using a specifically designed graph representation. While most of the existing methods work with undirected graphs, the recent work of (Monti et al., 2018) propose an approach for processing data defined on a directed graph by exploiting local graph motifs that describe its connectivity patterns. The main differences of this method with our work are that first, this method assumes that the directed graph is already given. In our problem, building such a graph that is able to fully take advantage of the image projective geometry is one of the contributions. Second, the approach in (Monti et al., 2018) does not use the knowledge of the coordinate system associated with omnidirectional images, which we however use in our architecture, in order to define filter orientations. The list of the aforementioned works is by no means extensive, therefore we refer the interested reader to the survey (Bronstein et al., 2017), which summarizes many geometric deep learning methods in detail.

**Omnidirectional image processing.** The most typical way of dealing with images taken by omni-directional cameras is to apply standard image processing techniques directly on the equirectangular projection images, which is one of the most common representations for the omnidirectional im-ages (De Simone et al., 2016). However, due to the strong distortion effects introduced by the process of unwrapping of the projection surface to a plane, standard techniques lose much of their efficiency, as the appearance of the same object may change depending on its location in the im-age. To overcome this problem the recent work of Khasanova & Frossard (2017b) suggests using graph-based special convolutional filers to adapt to the geometry of omnidirectional images. This method, however, relies on the convolutional operation defined in the spectral domain, which leads

to isotropic filters and may reduce the complexity of trained filters. We, on the other hand, propose to build anisotropic graphs-based convolutional filters that do not have this limitation. A different direction is taken by the authors of (Su & Grauman, 2017) who suggest adapting the size of the convolutional kernel to the elevation of the equirectangular image. The main limitation of this technique, however, is that it requires a significantly larger number of parameters than the competing techniques, as it does not have the weight sharing property of CNNs. It rather requires learning different convolutional filters for different elevations of the equirectangular image. Further, Jeon & Kim (2017) propose to learn the shape of the convolutional filter, by learning the sampling locations (position offsets), where the elements of the filter are evaluated. The authors of (Dai et al., 2017) extend later this idea and suggest learning dynamic offsets of the filter elements depending on the image content, which allows the filters to adapt to different parts of the image. This method is quite flexible, however in the context of omnidirectional images requires an extensive training set, as the network needs to learn how it should react to various objects appearing at any possible elevation. In our work we rather take advantage of the knowledge of the image projective geometry and use this knowledge in the design of our architecture to adapt the size and shape of convolutional filters.

A different approach is suggested by Cohen et al. (2018), who introduces a CNN that is designed for spherical shapes and define filters directly on its surface. This method, however, is specifically designed for processing spherical images, while our approach is easily adapted to different kind of shapes, which we show in our experiments. Further, the methods of (Coors et al., 2018; Tateno et al., 2018) suggest a different way of compensating for the distortion of omnidirectional image. They suggest adapting the sampling locations of the convolutional filters to the geometry of the lens by projecting kernels to the sphere and using interpolated pixel values on the projected locations for implementing the convolutional filters. While these works are the closest to in spirit to ours, we propose a more general architecture, which permits to adapt the shape and size of the convolutional kernel to the location of omnidirectional image, and therefore to use the information about all the pixels and not only of a subset of them.

Then, the authors in (Monroy et al., 2018; Ruder et al., 2018) suggest a completely different approach to tackle the distortion of omnidirectional images. Instead of working with equirectangular images, they propose to project an omnidirectional image to a cube, where each of its faces represents an image that would have been seen through a regular perspective camera, with the optical center located in the center of the cube (Chen et al., 2018). Representing an omnidirectional image in this way allows having less noticeable distortion effects as compared to equirectangular images. This representation, however, suffers from another type of distortion that appears due to discontinuity effect on the borders between the faces of the cube. To mitigate this issue, Monroy et al. (2018) propose to apply a smoothing filter as a post-processing step and Ruder et al. (2018) suggest an algorithm that enforces consistency between the neighboring facets of the cube. Contrary to the mentioned approaches, as we model cube surface as a graph, our algorithm can easily handle the discontinuity problem and adapt to image distortions introduced by the cube-map projection.

## 3 GEOMETRY-AWARE CNN

In this section we describe our algorithm, which adapts convolutional filters to the distortion of omnidirectional images. We start with the introduction of the equirectangular projection, as it is one of common ways to represent images from the omnidirecitonal cameras (De Simone et al., 2016; Coors et al., 2018). We then describe our graph-based representation learning framework.

### 3.1 EQUIRECTANGULAR PROJECTION

Omnidirectional visual content can be represented as a sphere with radius $r$, where the user is assumed to be located at the center. Each 3D point can be projected to a point on the surface of this sphere, which is described by spherical coordinates, namely a longitude $\theta \in [-\pi, \pi]$ and a latitude $\phi \in [-\frac{\pi}{2}, \frac{\pi}{2}]$.

Omnidirectional images are generally not processed directly in their native geometry, but they are first projected to the 2D plane where classical image processing techniques can be activated. One of the popular projections involves sampling the data on the sphere with equal steps $\Delta\theta$ and $\Delta\phi$, which results in an equrectangular image on the 2D plane. Thus, each point of equrectangular image

is defined by its spherical coordinates. To describe this projection let us introduce the tangent plane $T$, which is tangent to the sphere in the point $(\theta_0, \phi_0)$. Thus, each point $(x, y)$ on the tangent plane is projected to the sphere surface $(\theta, \phi)$ as follows (Coors et al., 2018):

$$
\begin{aligned}
\phi(x, y) &= \sin^{-1}(\cos \nu \sin \phi_0 + \tfrac{y \sin \nu \cos \phi_0}{\rho}) \\
\theta(x, y) &= \theta_0 + \tan^{-1}(\tfrac{x \sin \nu}{\rho \cos \phi_0 \cos \nu - y \sin \phi_0 \sin \nu})
\end{aligned} \quad .
\tag{1}
$$

where $\rho = \sqrt{(x^2 + y^2)}$ and $\nu = \tan^{-1} \rho$.

In order to have similar filter response regardless of the position of the object we model distortion of the applied filter. Thus, similarly to the works of (Khasanova & Frossard, 2017b) and (Coors et al., 2018), we define the filter kernel on this tangent plane $T$. Fig. 1 illustrates a sample equirectangular image with various kernels corresponding to tangent plane at different positions on the sphere. As we can see the projected area is different for various tangent plane locations. This projected area defines the support of our geometry-aware features, as described in the next section.

## 3.2 GEOMETRY-ADAPTIVE FILTERS

In the context of omnidirectional cameras, the main drawback of the classical convolutional approach is that it applies the same rectangular filters to different image positions. However, as we mentioned before, equirectangular images have different statistics at various elevations. Therefore, we propose to adapt the size and shape of the filter to the elevation on the spherical image.

To do so we propose to use a graph-based approach, which has recently become popular. The nodes of this graph $v_i \in \mathcal{G}$ and signal $y(v_i)$ defined on the nodes represent pixels and their intensity values respectively. Based on this graph $\mathcal{G}$ we can then use the Laplacian polynomial filters $\mathcal{F}$ as proposed in (Defferrard et al., 2016; Khasanova & Frossard, 2017a), where normalized Laplacian operator is defined as follows:

$$
\mathcal{L} = I - D^{-1/2} \mathcal{A} D^{-1/2},
\tag{2}
$$

where $I$ is an identity matrix, $D$ is a diagonal degree matrix and $\mathcal{A}$ is an adjacency matrix, which for each node $v_p$ of graph $\mathcal{G}$ defines its neighborhood $N_p$. Then, $\mathcal{F}$ has the following form:

$$
\mathcal{F} = \sum_{l=1}^{M} \alpha_l \mathcal{L}^l,
\tag{3}
$$

where the $\alpha_l$ are the trainable parameters, $\mathcal{L}$ is a Laplacian matrix and $M$ is the degree of $\mathcal{F}$.

Using these filters we can construct a Deep learning architecture and use it for various tasks, e.g., classification of omnidirectional images. The main advantage of our approach is that by appropriately constructing the graph $\mathcal{G}$ we make the Laplacian polynomial filters $\mathcal{F}$ react similarly to the same object seen at different elevation of the equirectangular image regardless of geometric distortion due to the spherical geometry. Here we call those filters geometry-aware (GA).

In order to adapt the GA-filter $\mathcal{F}$ to the elevation we build a graph $\mathcal{G}$ in such a way that the neighbourhood of each node is different for different elevations of the omnidirectional image. In the following section we describe two approaches that rely on undirected and directed graphs respectively, which consequently define isotropic and anisotropic filters. Then we describe in more details the polynomial anisotropic filter $\mathcal{F}$ that is used for directed graphs..

**Undirected graph construction for adaptive filtering.** To adapt $\mathcal{F}$ to the elevation level we construct a graph $\mathcal{G}$ with nodes that have different neighborhoods depending on the elevation. To do so we define a circular area on a tangent plane $\mathcal{T}$, centered in the tangency point. Then we move $\mathcal{T}$ such that it becomes tangent to the sphere $\mathcal{S}$ in different positions $v_p$ and project the circular area onto $\mathcal{S}$. For every point of $\mathcal{S}$ this creates a neighborhood $N_p$, which changes its shape and size together with the elevation, as can be seen in Fig. 1.

Based on this geometry adaptive neighborhood, we then construct the graph $\mathcal{G}$ in the following way. We connect the node $v_p \in \mathcal{G}$, corresponding to a tangent point on the sphere, with the node $v_j \in N_p$. The corresponding edge $e_{pi}$ has a weight $w_{pi}$ that is inversely proportional to the Euclidean distance between $v_p$ and $v_i$, which are defined on the sphere:

$$
\begin{aligned}
w_{pi} &= ||v_p - v_i||_{L_2}^{-1}, \quad v_i \in N_p, \\
w_{pi} &= 0, \quad\quad\quad\quad v_i \notin N_p.
\end{aligned}
\tag{4}
$$

This allows us to vary the size of the neighbourhood according to the geometry of the omnidirectional image for each node in $\mathcal{G}$, and weight the contribution of the nodes to the final filter response according to their distances to $v_p$. Therefore, depending on the elevation the filter is changing its shape and size.

While effective, filter $\mathcal{F}$ does not have a defined orientation in space as according to Eq. (3) the filter applies the same weights $\alpha_l$ to all nodes in the l-hoop neighborhood, with the contribution of each node being weighted by the distance to $v_p$. This results in $\mathcal{F}$ being isotropic, which leads to suboptimal representation, as the network is not able to encode the orientation of the object.

**Directed graphs construction.** In order to overcome the limitations of isotropic filters, we propose to replace a single undirected graph $\mathcal{G}$ with multiple directed graphs $\mathcal{G}_k$, where each $\mathcal{G}_k$ defines its own orientation. Let us consider the case of a 3x3 classical convolutional filter. In this case the filter has 9 distinct elements. To mimic the same structure with our graph convolutional filters we employ the following algorithm. First we define 9 non-overlapping areas $s_k, k = 1..9$ on the tangent plane, which together form a rectangular region that is centered in the tangency point $v_p$ of $T$ as defined in the Fig. 2. This rectangular region effectively defines the receptive field of the filter on the tangent plane. Then we build a set of nine directed graphs $G_k, k = 1..9$ in the similar way, as mentioned in the previous section for undirected graph. In particular in order to build graph $\mathcal{G}_k$ we do as follows. For the area $s_k$ and for every node $v_p$ we move the tangent plane at point $v_p$ and then project $s_k$ from $T$ onto the sphere. This operation defines a specific neighborhood $N_k(p)$ on the sphere that consists of the points that belong to the projection of the region $s_k$ from the plane $T$. We then connect $v_p$ with a directed edge to each of these points, where the weight of the edge is defined in Eq. (4). Note that the direction of the edge is very important, because connecting $v_p$ and $v_i$ with an undirected edge forces $v_p$ to be part of the neighborhood $N_k(i)$. This, however, is not possible, as the neighborhood $N_k(i)$ is computed by projecting the area $s_k$ from the plane $T$ that is tangent to the sphere at point $v_i$ and does not include $v_p$.

This results in construction of the directed graph $\mathcal{G}_k$, which corresponds to the $k^{th}$ region of the filter, illustrated in Fig. 2. We repeat this operation for all the areas $s_k, k = 1..9$ of our filter, which leads to creation of 9 directed graphs $\mathcal{G}_k, k = 1..9$. Given this set of graphs $\mathcal{G}_k$ we define the resulting convolutional operation $F$ as follows:

$$F = \sum_{k=1}^{9} \mathcal{F}_k, \tag{5}$$

where $\mathcal{F}_k$ is the filtering operation defined on the graph $\mathcal{G}_k$. Note that this filtering operation is slightly different from the operation that is used when working with undirected graphs and is discussed in more details in the following section.

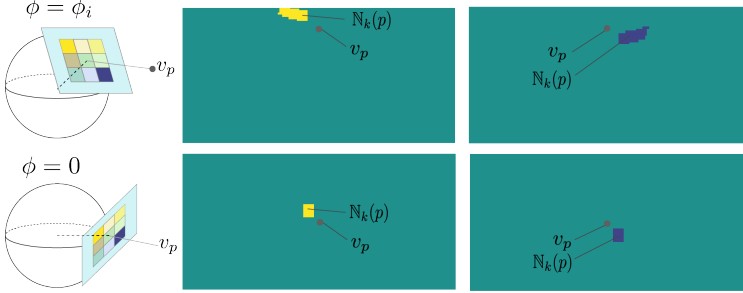

Figure 2: Illustration of $3 \times 3$ GA-filter kernel, defined on the tangent plane. Projection of area $s_k$ forms the neighborhood $N_k(p)$ of the node $v_p$, where the GA-filter is applied. The right part of the figure illustrates the change of the neighborhood $N_k(p)$ depending on the location of $v_p$ and the chosen filter area $s_k$.

To sum up, the introduced graph construction process allows having anisotropic filters $F$, defined in Eq. (5) that are capable of capturing the orientation of the object and therefore learn more meaningful feature representation for an image compared to the isotropic graph-based filters. It is important

to note that in this paper we use the set of 9 non-overlapping rectangular areas defined on the tangent plane, as shown by Fig. 2, due to their rough correspondence to the elements of a $3 \times 3$ convolutional filter. However, our method can be easily extended to an arbitrary number of such areas with arbitrary shapes.

**Geometry aware anisotropic filters.**   For directed graphs $\mathcal{G}_k$ Laplacian matrix is not defined, therefore, we use the polynomial filters proposed in (Sakiyama et al., 2017). Instead of the Laplacian matrix these filters rely on the normalized adjacency matrix $\mathcal{A}_k$, which is defined as follows:

$$\mathcal{A}_k = D_k^{-1} A_k, \tag{6}$$

where $A_k$ and $D_k$ are the weighted adjacency and the diagonal degree matrices of graph $\mathcal{G}_k$ respectively. The elements of $D_k$ are computed as $D_k(m, m) = \sum_n A_k(m, n)$. Then, we define filters in the following way:

$$\mathcal{F}_k = \alpha_0^{(k)} + \alpha_1^{(k)} \mathcal{A}_k, \tag{7}$$

where $\alpha_0^{(k)}, \alpha_1^{(k)}$ are the training parameters of our filter. Here, we use polynomial filters of degree 1, as they achieve good balance between speed and performance.

**Network architecture.**   The introduced approach focuses on the modification of the convolutional layer to incorporate the knowledge about the image projective geometry inside the neural network. Thus, it can be applied for a broad variety of tasks. In this chapter we focus on the image classification and compression problems. For the former one we use a relatively standard architecture that consists of a sequence of convolutional layers with the introduced graph-based filters, followed by a sequence of the fully connected layers. For the compression task we use the architecture proposed in (Ballé et al., 2017) and replace its convolutional filters with the proposed graph-based ones.

**Discussion.**   Our method can be seen as a generalization of different approaches that have been developed for omnidirectional images. For example, if the node $v_p$ at elevation $\phi_i$ has only one neighbor in each direction and the weight of the edges between nodes is always equal to one, it will be the standard CNN method (LeCun et al., 2001). Further, if these neighbors correspond to the projected points it becomes the recently proposed algorithm of (Coors et al., 2018). Finally, if we replace directed graphs with a single undirected one we get the same behavior of the polynomial filters as described in graph-based deep learning methods (Khasanova & Frossard, 2017a; Defferrard et al., 2016; Kipf & Welling, 2017; Khasanova & Frossard, 2017b).

## 4 RESULTS

In this section we illustrate the performance of our approach. We start by evaluating our method with respect to competing approaches on the task of classification images that are projected to different surfaces. Finally, to show the generality of our approach and illustrate the effectiveness of the anisotropic graph-based filters, we evaluate our method on the image compression task.

### 4.1 IMAGE CLASSIFICATION

In this section we first introduce the datasets that we used for the evaluation of our method. We then discuss the baseline approaches and architectures, which we use in our experiments. Finally, we show the quantitative comparison of our method with the competing ones.

**Datasets.**   We evaluate our method on three different types of data, which feature different surface geometries, where the images are projected:

- **Spherical** dataset (S) consists of images projected on different locations on a sphere. The resulting spherical images are then unwrapped to for equirectangular images, as described in Section 3.1.
- **Mod-spherical** dataset (MS) features image projections on more complicated surfaces that are depicted by Fig. 3 together with the representative examples of projected images. This dataset itself consists of three different versions: MS1, MS2, MS3 which correspond to the

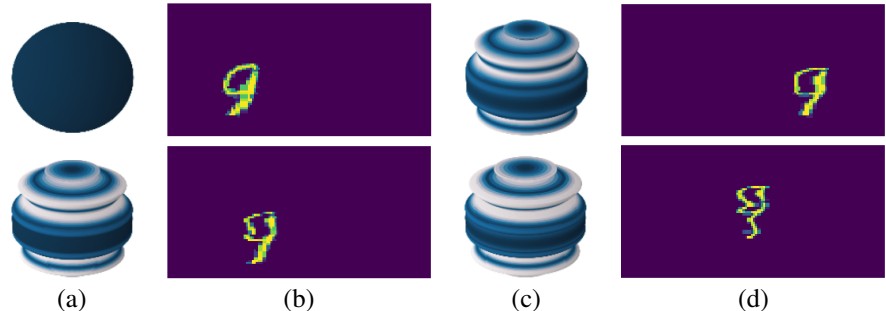

Figure 3: Illustration of surfaces (a,c) and corresponding equirectangular images (b,d) of the digit 9 projected to these surfaces at random positions. White color of surfaces highlights the furthest points to the spherical surface of the same radius in terms of Euclidean distance; blue color indicates the closest points.

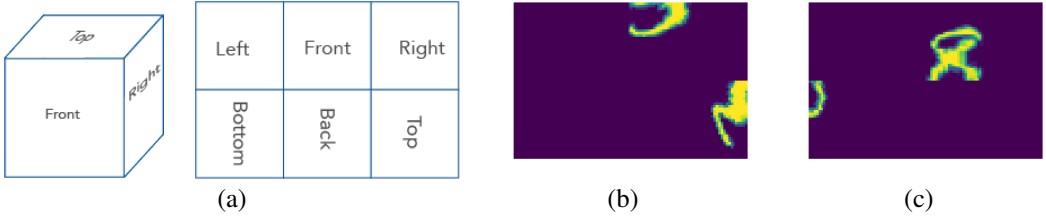

Figure 4: **Cube-map projection**: (a) schematic illustration of the unwrapping process of the cube surface with a baseball arrangement (Chen et al., 2018) onto a planar surface; (b,c) sample projections of images from the MNIST dataset on the cube surface unwrapped into rectangular images.

surfaces which are getting further away from the spherical one. A more detailed discussion about the type of projection and the surface geometry used in these datasets can be found in ApendixB.

- **Fish-eye** dataset (F) consists of images projected on different locations on a sphere using stereographic projection Bettonvil (2005), which is frequently used in fish-eye cameras.

- **Cube-map** dataset (CM) features projection of the images on the cube as shown by Fig. 4. This type of projection has recently gained popularity for handling omnidirectional images, due to its ability to reduce distortion artifacts that appear due to the spherical geometry.

In all these dataset we use MNIST images from (LeCun & Cortes, 2010) , which are divided into train, validation and test sets with 54k, 6k and 10k samples respectively.

**Architecture.** We compare our approach with standard ConvNets, the algorithm proposed in (Cohen et al., 2018) and other graph-based methods. We present our result in the Table 1[1]. For the graph-based methods we investigate 3 possible way of constructing $\mathcal{G}$:

- Regular grid-graph with 8 neighbors and all equal weights $w_{ij} = 1$;

- Regular grid-graph with 8 neighbors and weights that depend on the Euclidean distance $d_{ij}$ between the nodes, as proposed in (Khasanova & Frossard, 2017b) $w_{ij} = d_{ij}^{-1}$;

- Irregular GA-graph $w_{ij} = d_{ij}^{-1}$ (isotropic filters from Section 3.2).

For all of them we build a normalized Laplacian matrix (Khasanova & Frossard, 2017b) and use polynomial filters of degree 1, which is equivalent to using $3 \times 3$ convolutional kernels. Therefore, for the standard ConvNet we similarly rely on filters of size $3 \times 3$.

---

[1]We were unable to compare our method to the recent work of (Coors et al., 2018) as to the best of our knowledge, there is no publicly available implementation.

Table 1: Evaluation of different approaches on Spherical (S), Mod-Spherical (MS1, MS2, MS2), Fish-eye (F) and Cube-Map (CM) datasets.

| Method | S | MS1 | MS2 | MS3 | F | CM |
|---|---|---|---|---|---|---|
| regular graph ($w_{ij} = 1$) | 69.4 | 64.3 | 64.1 | 62.8 | 71.8 | 40.0 |
| regular graph ($w_{ij} = 1/d_{ij}$) | 69.8 | 63.4 | 64.5 | 62.5 | 70.2 | 40.5 |
| GA graph ($w_{ij} = 1/d_{ij}$) | 70.2 | 63.9 | 62.5 | 62.8 | 72.1 | 44.2 |
| ConvNets | 94.2 | 91.3 | 91.2 | 90.5 | 93.4 | 79.4 |
| (Cohen et al., 2018) | 95.2 | 84.5 | 83.3 | 80.9 | 94.9 | – |
| Ours | **96.9** | **95.1** | **95.3** | **94.9** | **95.7** | **84.3** |

All the competing approaches use the networks of roughly the same complexity. For all the methods we use the architectures of similar structure and roughly the same number of parameters. For all the graph-based approaches we use the graph-based convolutions with stride two on each layer, which in turn requires building graph for each new layer according to its respective sampling. The exact architecture of the classification network is illustrated in Appendix A. For the method of (Cohen et al., 2018) we used the architecture proposed in the paper with roughly the same number of parameters as in competing approaches.

**Evaluation.** We present the result of comparison of our approach with the baseline methods in Table 1. Our method significantly outperforms the standard ConvNets, as it is designed to use the geometry of the omnidirectional images. Further it shows a much higher accuracy then other graph-based techniques, which rely on isotropic filters. Further, our method achieves comparably accuracy with (Cohen et al., 2018) on spherical image representation, however, our method is more general, therefore, it outperforms (Cohen et al., 2018) on other datasets. Finally, we are able to run our approach on cub-map projection while the SphericalCNN by design is not applicable to such kind of images.

## 4.2 IMAGE COMPRESSION

In all our previous experiment we have focused on evaluating our approach on the image classification task. To show the generality of our method and better illustrate the effectiveness of anisotropic graph-based filters, we now evaluate their performance on an image compression problem. For this task, we choose to modify the architecture introduced in (Ballé et al., 2017) by replacing the ordinary convolutional layers with our own graph-based convolutions. In this section we first introduce our approach and then compare the performance of the two graph-based methods, which rely on isotropic and anisotropic graph-based filters respectively.

**Image compression framework.** The method introduced in (Ballé et al., 2017), presents the process of image compression is an optimization of the tradeoff between having small distortion of the pixel intensity values and the small number of bits that are required for storing the compressed representation of these values. As described in (Ballé et al., 2016; Ballé et al., 2017), this optimization can be represented as a variational autoencoder. For more details we refer to Appendix C. In the context of omnidirectional images, we propose to modify the method proposed in (Ballé et al., 2017) by using our geometry-aware filters instead of standard convolutional ones.

**Evaluation.** We now evaluate the performance of our approach. For this experiment we have implemented two versions of the system. One with isotropic graph-based filters and the other one with the anisotropic ones. We further evaluate the original method (Ballé et al., 2017) for the sake of completeness. All three methods are trained and tested on the same splits of the modified version of the dataset (Xiao et al., 2012), which consists of omnidirectional images projected onto a cube. From this dataset we use 3900 images for training and 1000 for testing of our approaches.

We compare the methods in terms of the Peak Signal to Noise Ratio (PSNR) with respect to the average number of bits per pixel (bpp). The results of the evaluation are presented in Fig. 5. As we can see, our method with anisotropic filters and (Ballé et al., 2017) show similar PSNR values and

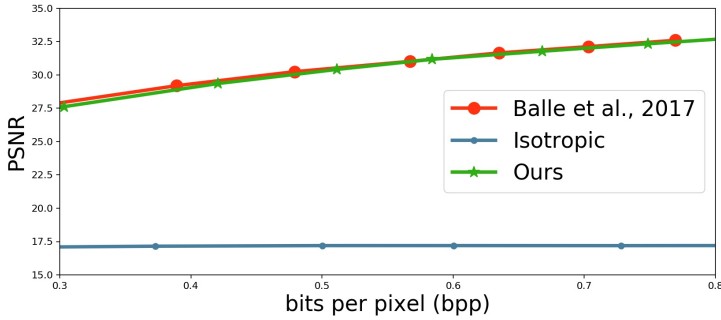

Figure 5: PSNR results of the decompressed images for different filter types with respect to bit per pixel values.

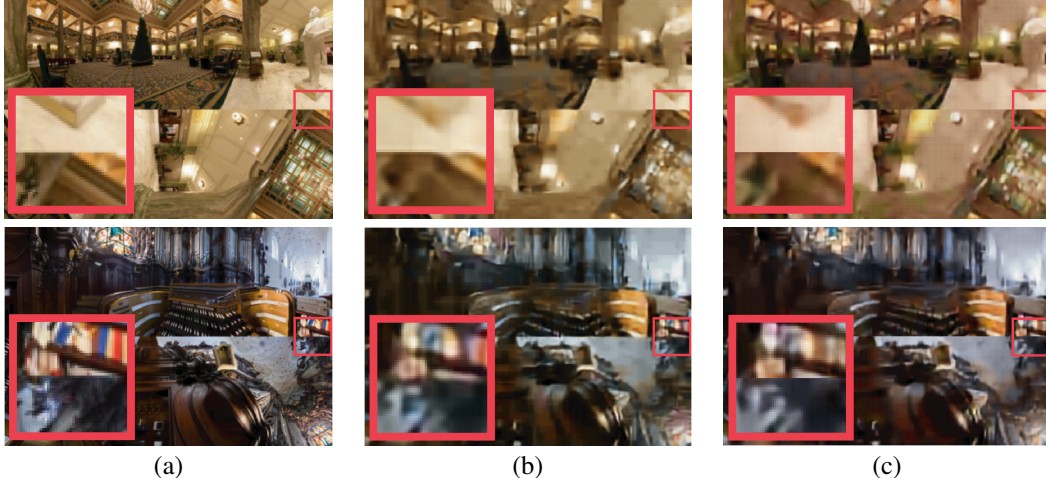

(a)                                  (b)                                  (c)

Figure 6: **Cube-map projection's artifacts** appear due to the discontinuity between unwrapped face's borders. On the left corners of original (a) and decompressed images (b,c) we show zoomed version of the image patch that illustrates the borders of the cube faces. The decompressed results (c), obtained by our proposed anisotropic geometry-aware filters does not smooth border of the faces as our method has access to the information about cube-map geometry, while the result (b) obtained with convolutional filter from (Ballé et al., 2017) smooths these borders. This smoothing can lead to perceptutally unpleasant result in various applications (e.g., virtual reality). (best seen in color)

significantly outperform the architecture with isotropic filters. Further, due to the fact that PSNR depends on the average difference in pixel values between the compressed image and the original one it is not able to reliably detect small artifacts that appear in the cube-map images, which are noticeable for humans. These artifacts are clearly seen in Fig. 6, where we illustrate that due to the knowledge about the image projective geometry, our approach correctly reconstructs the areas along cube borders, while the method of Ballé et al. (2017) over-smooths these areas.

## 5   CONCLUSION

In this paper we have presented generic way of graph construction that allows incorporating the information about the image geometry inside the neural network. Further, we have introduced the graph-based geometry aware convolutional filters that adapt their shape and size to the geometry of the projection surface. In contrast to many existing graph-based filters, our filters are anisotropic, which allows to better adjust to the specific properties of the problem. Our illustrative experiments show state-of-the-art performance of our approach applied to image classification and compression tasks in the presence of various types of image distortions.

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

APPENDIX

## A    ARCHITECTURE OF CLASSIFICATION FRAMEWORK

Fig. 7 illustrates the architecture of our classification network, which we use in Section 4.1.

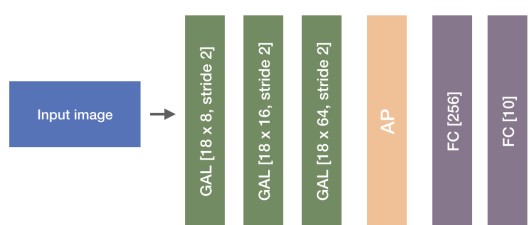

Figure 7: Our classification network consists of three geometry-aware graph-based convolutional layers (GAL) with stride 2. Each of the GAL [$X \times Y$, stride 2] convolutional layers contains $Y$ filters with $X$ parameters. The convolutional layers are followed by an average pooling operation AP and two fully-connected layers FC[$Z$] with $Z$ neurons each.

## B    MODIFIED SPHERICAL SURFACE.

In order to evaluate the performance of our method as a function of deformation of the spherical surface, we have created a set of datasets by projecting the MNIST images to random locations of the surfaces, which have shapes, shown in Fig. 3 (a,c) and unwrap these spherical images to equirectangular ones. The white color in Fig. 3 (a,c) denotes the areas of the generated surface that are the furthest from the spherical surface of the same radius. The Fig. 3 (b,d) illustrates sample images of digits projected onto the respective surfaces. Each of the aforementioned surfaces is the following modification of a spherical one from Eq. (1):

$$
\begin{aligned}
x &= \cos(\phi_i)\sin(\theta_i - \theta_0) \\
y &= (\cos(\phi_0)\sin(\phi_i + p(\phi_i, r, l)) - \sin(\phi_0 + p(\phi_0, r, l))\cos(\phi_i)\cos(\theta_i - \theta_0))/c, \\
c &= \sin(\phi_i + p(\phi_i, r, l))\sin(\phi_0 + p(\phi_0, r, l)) + \cos(\phi_i)\cos(\phi_0)\cos(\theta_i - \theta_0),
\end{aligned}
\tag{8}
$$

where $(x, y)$ are the coordinates on the tangent plane and $p(\phi, r, l)$ is the perturbation function that can be written as

$$
p(\phi, r, l) = r\sin^{-1}(\sin(l\phi)),
\tag{9}
$$

where $\phi$ is the elevation level; $r$ is the parameter that regulates the perturbation magnitude and $l$ defines frequency of the perturbation signal. In our experiments we have set $l = 10$. Note that for a specific case of $r = 0$ we get the ordinary spherical surface. We then use Eq. 8 to construct the graph $\mathcal{G}$ that allows our method to adapt to the surface geometry and evaluate our method on each of the generated datasets.

## C    COMPRESSION

In this section we briefly describe compression approach, which proposed by **?**. An input image $x$ is encoded using a function $g_a(x; \alpha)$, which results in the respective latent representation $y$. Then, $y$ is quantized into $\hat{y}$, which can be losslessly compressed using entropy coding algorithms. This $\hat{y}$ is then passed to the decoder $g_s(\hat{y}; \beta)$ at the decompression step, which results in a decompressed image $\hat{x}$. Here, we denote by $\alpha$ and $\beta$ the parameters of the encoding and decoding algorithms respectively. While both encoder and decoder can be represented as a differentiable function, the process of quantization is non-differentiable. Therefore the authors of (Ballé et al., 2016) propose to replace quantization with an additive uniform noise at the training step as follows:

$$
\tilde{y}_i = y + \Delta y,
\tag{10}
$$

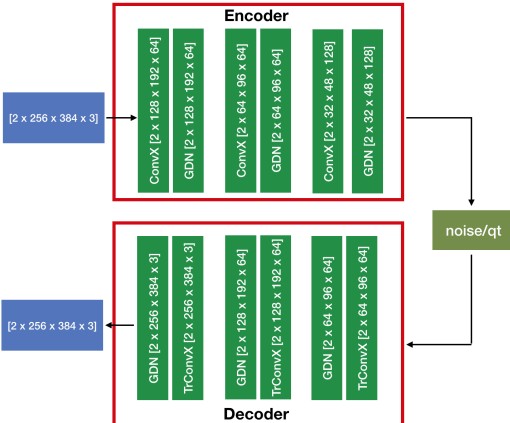

Figure 8: **Architecture of compression algorithm**. For our experiment we use omnidirectional images projected to the cube surface with a baseball arrangement with size $[256 \times 384 \times 3]$ grouped in batches of size 2. Here, ConvX and TrConvX denote convolutional and transpose convolutional layers with stride 2, where X corresponds to the possible choice of either standard convolutional or proposed geometry-aware graph-based filters; and GDN is a normalization layer which is proposed by (Ballé et al., 2016).

where $\Delta y$ denotes additive i.i.d uniform noise. This trick allows to perform the end-to-end optimization of both the encoder and decoder parameters using the following loss function:

$$L(\alpha, \beta) = E_{x, \Delta y}\left[ -\sum_i log_2 p_{\tilde{y}_i}(g_a(x; \alpha) + \Delta y) + \lambda d(g_s(g_a(x; \alpha) + \Delta y; \beta), x)\right], \quad (11)$$

where $g_s, g_a$ are convolutional deep neural networks, $d$ represents the distance between the images and $\lambda$ is a weighting parameter. Thus, during the training step, we add noise (according to Eq. (10)) to be able to back propagate the error and at the inference time we apply quantization to the latent representation $y$. The overall architecture that we use is similar to the one proposed in (Ballé et al., 2017) and is summarized in Fig. 8. Further, the method of (Ballé et al., 2017) relies on the standard convolutional layers, which are practical for ordinary images: they allow learning local image structures independently of their location in the image.

**Additional visual results.** We run experiment with described compression architecture, where we compare three approaches: original and methods, where we replace convolutional filter from the architecture to graph-based isotropic and geometry-aware filters. Fig. 9 further illustrates some visual comparison of the methods and we can see isotropic filters produce over-smoothed decompressed images, which do not look realistic and result in very low PSNR values. On the other hand our method with anisotropic filters is able to produce sharp results.

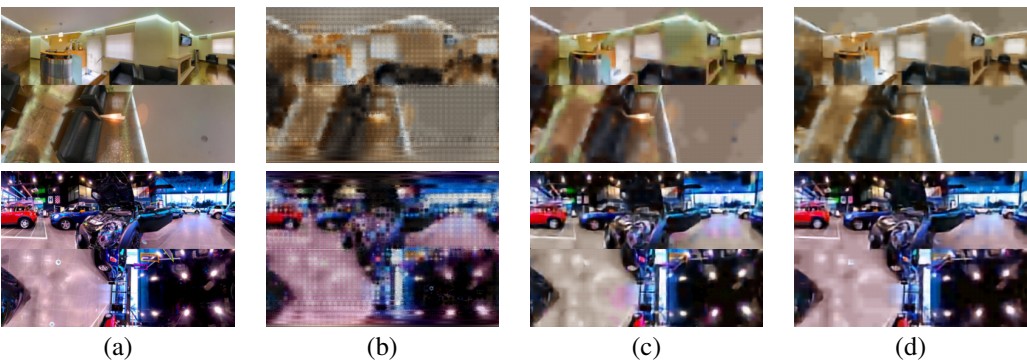

(a)  (b)  (c)  (d)

Figure 9: **Decompression result**: original images (a); decompressed image obtained by our algorithm with isotropic geometry-aware (b), proposed anisotropic geometry-aware (c) and convolutional filters from (Ballé et al., 2017) (d). (best seen in colors)

