# OpenReview forum: "Geometry aware convolutional filters for omnidirectional images representation"
_ICLR.cc/2019/Conference_

### Official Review · AnonReviewer2 · 2018-10-21

**Rating:** 4
**Confidence:** 5

**Review:**

The paper proposes a new way of defining CNNs for omnidirectional images. The method is based on graph convolutional networks, and in contrast to previous work, is applicable to other geometries than spherical ones (e.g. fisheye cameras). Since standard graph CNNs are unable to tell left from right (and up from down, etc.), a key question is how to define anisotropic filters. This is achieved by introducing several directed graphs that have orientation built into the graph structure.

The paper is fairly well written, and contains some new ideas. However, the method seems ad-hoc, somewhat difficult to implement, and numerically brittle. Moreover, the method is not equivariant to rotations, and no other justification is given for why it makes sense to stack the proposed layers to form a multi-layer network.

The results are underwhelming. Only experiments with small networks on MNIST variants are presented. A very marginal improvement over SphericalCNNs is demonstrated on spherical MNIST. I'm confused by the dataset used: The authors write that they created their own spherical MNIST dataset, which will be made publicly available as a contribution of the paper. However, although the present paper fails to mention it, Cohen et al. also released such a dataset [1], which raises the question for why a new one is needed and whether this is really a useful contribution or only results in more difficulty comparing results. Also, it is not stated whether the 95.2 result for SphericalCNNs was obtained from the authors' dataset or from [1]. If the latter, the numbers are not comparable.

The first part of section 3.2 is not very clear. For example, L^l is not defined. L is called the Laplacian matrix, but the Laplacian is not defined. It would be better to make this section more self contained.

In the related work section, it is stated that Cohen et al. use isotropic filters, but this is not correct. In the first layer they use general oriented spherical filters, and in later layers they use SO(3) filters, which allows anisotropy in every layer. Estevez et al. [2] do use isotropic spherical filters.

In principle, the method is applicable to different geometries than the spherical one. However, this ability is only demonstrated on artificial distortions of a sphere (fig 3), not practically relevant geometries like those found fisheye lenses.

In summary, since the approach seems a bit un-principled, does not have nice theoretical properties, and the results are not convincing, I recommend against acceptance of this paper in its current form.


[1] https://github.com/jonas-koehler/s2cnn/tree/master/examples/mnist
[2] Estevez et al. Learning SO(3) Equivariant Representations with Spherical CNNs

---

> ### Author Response · Authors · 2018-11-26
> **Re: Review**
>
> We would like to thank you for the comments. We have updated the description of the approach to make it more clear and self contained. We have also added a better description of the architecture that we use and reorganized the Section 4 to better summarize the results. Please find a detailed answer to the raised questions below. For simplicity we organized our response as a list of answers, one for each paragraph in the reviewer’s comment.
>
> 1. Our approach is designed with the goal of incorporating the available prior knowledge inside the optimization procedure. Unlike the existing techniques, such as SphericalCNN, which work very well for spherical surfaces, our method is able to adapt its filters to any type of surface geometry as we have shown in our experiments, which proves the generality of the approach. During our experiments we found the optimization to be stable to different choices of hyperparameters. While it is true that our filters are not equivariant to rotations it is important for our network to have multiple layers as this allows the neurons of the deeper layers to have larger receptive field. This increase happens, because we apply our convolutional filters with stride 2, we have explained this better in the revised version of the paper.
>
> 2. We have reorganized the results section of the paper to better illustrate the performance of our approach in comparison to the competing techniques. Briefly, regarding the dataset, we apologize for the confusion. We are aware of the dataset of omnidirectional images that is presented in the SphericalCNN paper. However, the dataset that we will make available is slightly different, as it consists of several different datasets that are obtained from MNIST by projecting its images on the surface of different geometric shapes (including simple spheres, modified spheres with different amount of added noise and cubes) and therefore permits developing techniques that are able to adapt to either one of the presented distortions or to all of them. Further, in our experiments we used our dataset (with identical training and test splits) for the evaluation of all the methods, including the SphericalCNN.
>
> 3. Thank you for this comment, in the original version, we indeed omitted the introduction of the Laplacian matrix and left only the reference to the literature due to the space limitations. We discuss it in more details in the updated version of the paper in Section 3.2.
>
> 4. Thank you for pointing out that the description of the work of Cohen et al. (2018) is not precise. We have updated Section 2 accordingly in the revised version of the paper.
>
> 5. Apart from the modified spherical geometries that we introduced to illustrate the generality of the approach, we have also shown the performance of our method for the case of cube-map projection of omnidirectional images. This type of projection has recently become popular and is frequently used for various task including compression. Further, in the revised version of the paper we have added the experiments with the stereographic projection, which is frequently used in fish-eye cameras. Finally, in the revised version we evaluate the performance of our method on a completely different problem of image compression, which shows the generality of our approach, as compared to the competing methods, which were developed solely for the classification task.
>
> 5. Finally, we believe that our approach provides with a generic way of dealing with different surface geometries as it is able to adapt the filter size and shape by only requiring the knowledge of the projection equations. This is a considerable advantage in comparison to other techniques that are designed only for specific type of surface geometry, as it has a wider application area. Further our suggested approach of defining the filter on the tangent plane is highly beneficial as it allows to avoid complex derivation, which will be required in case analytical computation of the changes in the size and shape of the filter. We have clarified these points in the revised version of the paper and modified the results section to better illustrate the advantages of the proposed technique.

---

### Official Review · AnonReviewer3 · 2018-11-03
**Interesting idea but needs better illustration**

**Rating:** 6
**Confidence:** 4

**Review:**

The paper introduces geometry-aware filters based on constructed graphs into the standard CNN for omnidirectional image classification. Overall, the idea is interesting and the authors propose an extrinsic way to respect the underlying geometry by using tangent space projection. Understanding the graph construction and filter definition is not easy from the text description. It would be better to use a figure to illustrate them.

1) How to define the size of the circular area on the tangent plane?

2) Will the filter change greatly with the definition of the weight function in the neighborhood? Since the point locates on the sphere, why not using the geodesic distance instead of the Euclidean distance?

3) It would be better to directly define the filter on the sphere and make it be intrinsic. The same filter on the tangent space may cover different sizes of regions on the sphere; while we prefer the filter has consistent coverage on the sphere.

4) The paper misses the discussion and comparison to Anisotropic CNN (ACNN) and mixture model network (moNet).

---

> ### Author Response · Authors · 2018-11-26
> **Re: Interesting idea but needs better illustration**
>
> Dear reviewer AnonReviewer3, thank you for your feedback. We have update the text of the paper to improve the clarity and also reorganized the Results section of the paper to better show the evaluation of our approach. Regarding your question we have the following clarifications:
>
> 1. To define an initial circular area of the filter we project a 3 x 3 pixels area from the equator of the sphere to the tangent plane. This gives the minimum radius for a filter such that when this filter is applied at different elevations on the sphere, each filter component corresponds to at least a point on the spherical image.
>
> 2. This is a good point. In principle as we sample sphere with a relatively high rate, the value of Euclidean and geodesic distances are very close to each other, so the performance of the network will not change much. In this work the weights are inversely proportional to Euclidean distance between the nodes. However, experimenting with the larger filter sizes, geodesic distances is an interesting direction for future research.
>
> 3. We apologize if is was not very clear from the text but our anisotropic filters are designed in a way that they cover the same area on the sphere independently of the elevation level. This in turn results in the areas of different sizes in the equirectangular representation of the spherical image. Further, even though for a spherical image we can have an analytic way of computing the size of the filter kernel to have the consistent coverage of the sphere, this approach has one drawback,that for any new type of projection this analytic equations should be re-derived. In our case, however, knowing the projection equations is enough to apply the method, which makes the approach readily available for working with different projection types.
>
> 4. Thank you for the reference, we added the discussion about Anisotropic CNN (ACNN) and mixture model network (moNet) to the Section 2 of the updated version of the paper. Briefly, the main difference between our approach and theirs is that in this work we propose a framework to build graphs to efficiently process equirectangular, cube mapping or another image projection, while the suggested references are designed for graphs (or manifolds) and do not rely on any prior information about geometrical structure of the projection.

---

### Official Review · AnonReviewer4 · 2018-11-10
**Experiments too limited to judge the merits**

**Rating:** 4
**Confidence:** 4

**Review:**

This paper proposed to use graph-based deep learning methods to apply deep learning techniques to images coming from omnidirectional cameras. It solves the problem of distorsions introduced by the projection of such images by replacing convolutions by graph-based convolutions, with in particular a combinaison of directed graphs which makes the network able to distinguish between orientations.

The paper is fairly well written and easy to follow, and the need for treating omnidirectional images differently is well motivated. However, since the novelty is not so much in the graph convolution method, or in the use of graph methods for treating spherical signals, but in the combined application of the particular graph method proposed to the domain of omnidirectional images, I would expect a more thorough experimental study of the merits of the method and architectural choices.

1. The projected MNIST dataset looks very localized on the sphere and therefore does not seem to leverage that much of the global connectivity of the graph, although it can integrate deformations. Since the dataset is manually projected, why not cover more of the sphere and allow for a more realistic setting with respect to omnidirectional images?
More generally, why not use a realistic high resolution classification dataset and project it on the sphere? While it wouldn't allow for all the characteristics of omnidirectional images such as the wrapping around at the borders, it would lead to a more challenging classification problem. Papers such as [Khasanova & Frossard, 2017a] have at least used two toy-like datasets to discuss the merits of their classification method (MNIST-012, ETH-80), and a direct comparison with these baselines is not offered in this work.

2. The method can be applied for a broad variety of tasks but by evaluating it in a classification setting only, it is difficult to have an estimate of its performance in a detection setting, where I would see more uses for the proposed methods in such settings (in particular with respect to rotationally invariant methods, which do not allow for localization).

3. I fail to see the relevance of the experiments in Section 4.2 for a realistic application. Supposing a good model for spherical deformations of a lens is known, what prevents one from computing a reasonable inverse mapping and mapping the images back to a sphere? If the mapping is non-invertible (overlaps), then at least using an approximate inverse mapping would yield a competitive baseline.
I am surprised at the loss of accuracy in Table 2 with respect to the spherical baseline. Can you identify the source of this loss? Did you retrain the networks for the different deformations, or did you only change the projection of the network trained on a sphere?

4. While the papers describes what happens at the level of the first filters, I did not find a clear explanation of what happens in upper layers, and find this point open to interpretation. Are graph convolutions used again based on the previous polynomial filter responses, sampling a bigger region on the sphere? Could you clarify this?

5. I would also like to see a study of the choice of the different scales used (in particular, size of the neighborhood).

Overall, I find that the paper introduces some interesting points but is too limited experimentally in its current form to allow for a fair evaluation of the merits of the method. Moreover, it leaves some important questions open as to how exactly it is applied (impact of sampling/neighborhood size, design of convolutions in upper layer...) which would need to be clarified and tested.

Additional small details:
- please do not use notation $\mathbb{N}_p$ for the neighborhood, it suggests integers
- p. 4 "While effective, these filters ... as according to Eq. (2) filter..." -> article missing for the word "filter"

---

> ### Author Response · Authors · 2018-11-26
> **Re: Experiments too limited to judge the merits**
>
> We thank reviewer for the comments. Please find the answer on your questions below:
>
> 1. We have experimented with the MNIST dataset to show the ability of our method adapt to different geometries of projective surface. We have further tried projecting other images on the sphere, but the resulting representations had various unrealistic artifacts on the borders of projected images. We therefore evaluated our method on a different image compression task, for which we could obtain good quality real omnidirectional images. We have updated the results section of our paper to include this experiment. The method further show that our approach is applicable to a wide range of tasks, while the competing method were solely designed for image classification.
>
> 2. This is a very good point. Indeed, as we discussed above, we have also experimented with the compression problem and show now the evaluation of our approach in Section 4.3 of the revised manuscript. Briefly we show that, due to the knowledge of the projective geometry that is encoded in the graph structure, we can easily avoid artifacts in the compressed images that are present when using conventional image coding methods.
>
> 3. Indeed for some of the surfaces it is possible to compute the mapping to the spherical one, however even if a reasonable mapping can be found, the necessity of computing one makes it harder to apply techniques developed for spherical images to other surfaces. While our method does not need this additional preprocessing step, which makes it readily applicable to different surfaces, as depicted by our experiments in the Results section. Further the fact that our approach works directly with the given surface allows it to avoid interpolation artifacts, which may be introduced during the interpolation process, when the given surface is mapped to the spherical one.
> In Table 2 we can see that our same algorithm adapts to different surface shapes as they are encoded in the graph structure. In these experiments, SphericalCNN does  not have the knowledge about the change of the projection, which results in a drop of performance. Regarding the second part of the question, we would like to clarify that we train separate networks for each of the surfaces.
>
> 4. Thank you for this question, we add the description about this point in Section 4.1 in updated version of the paper. For all the graph-based approaches we use the graph-based convolutions with stride two on each layer. This allows to increase the size of the receptive field for the neurons in the deeper layers of the network, similarly to the classic ConvNets. This in turns allows the network to process the information from the large area on the sphere without increasing the number of parameters. It is important to note here that having a strided convolution requires building a separate graph for every convolutional layer of the network, which is a subsampled version of the graph from the previous convolutional layer.
> 5. Thank you for the comment. In this paper we mostly focus on the introduction of the generic way of defining an anisotropic graph-based convolutional filter. We have experimented with some variations of shape and size of the filters, and did not observe significant changes in performance. Nevertheless, we believe that a complete study of all possible modifications of the of shape and size of the areas on the tangent plane that define the filter is a very interesting direction for the future research.
>
> We have modified the text of the paper to better focus on the advantages of the proposed technique. We have further added an evaluation of our approach on a different image compression problem, which shows the generality of the approach with respect to the competing methods.
>
> We would also like to further thank reviewer for the style suggestion and pointing out the typo. We updated the text accordingly.

---

### Public Comment · ~Michael_Bronstein1 · 2018-09-30
**previous works on geometry-aware deep learning**

The authors should be aware of a large amount of geometry-aware deep learning methods that are directly related to their work, especially in the intersection of learning, vision, and graphics. In [1], the first intrinsic CNN-like architecture was proposed for manifolds, further extended in [2-4]. In particular, in [2-3] anisotropic convolution filters on manifolds/meshes were proposed. In [6], anisotropic diffusion was extended to general graphs using graph motifs. These approaches can be considered as particular cases of the MoNet architecture [4], which in turn was extended in [5] using more general learnable local operators and dynamic graph updates.Finally, the authors may refer to a review paper [8] on geometric deep learning methods. I would be appropriate to compare to these methods, or at least discuss the differences from the proposed approach.

1. Geodesic convolutional neural networks on Riemannian manifolds, ICCV Workshops 2015.

2. Learning shape correspondence with anisotropic convolutional neural networks, NIPS 2016.

3. Anisotropic diffusion descriptors, Computer Graphics Forum 35(2):431-441, 2016.

4. Geometric deep learning on graphs and manifolds using mixture model CNNs, CVPR 2017.

5. Dynamic Graph CNN for learning on point clouds, arXiv:1712.00268

6. MotifNet: a motif-based Graph Convolutional Network for directed graphs, arXiv:1802.01572

7. Geometric deep learning: going beyond Euclidean data, IEEE Signal Processing Magazine, 34(4):18-42, 2017

---

> ### Author Response · Authors · 2018-10-07
> **Re: previous works on geometry-aware deep learning**
>
> Dear Michael,
>
> Thank you very much for the detailed comments. We are well aware of most of these works and their general relation to our proposal in terms of using geometry in the deep learning context. We, however, believe that these works focus on a different problem than ours, namely the one of 3D structure processing. In this paper, we rather propose a new approach to process omnidirectional images, where the challenge is to use the knowledge about image distortion in order to create effective representations. As deep networks suffer from interpretability, there is no straightforward way to incorporate this knowledge to the learning process. Therefore, our contribution is in suggesting an effective way of building several directed graphs to make the filters aware of the geometry of omnidirectional images. Of course, the sphere can be considered as a specific 3D structure - we rather argue that, if the geometry of images is known and fixed, it can be incorporated right away in the learning algorithm, instead of using generic learning solutions.
>
> Nevertheless, as this might lead to confusion, we will extend our related work by including a subsection about the relation between other geometrical graph-based approaches and our convolutional filters for omnidirectional images, We will specifically highlight the differences between these families of methods. Below we briefly summarise these differences.
>
> The method of [1] propose to define a patch around each point on a manifold using polar system of coordinates. Further, [2,3] propose different ways of constructing such patches on point clouds. The works [2,3] are in spirit similar to our approach in that they aim at creating anisotropic filters that operate on graphs. However, from the methodological perspective they are quite different, as they require computing eigendecomposition in order to model position- and direction-dependent filters, which is a time consuming process. The methods of [1,2,3] were generalised in the work [4], where the authors suggest defining a local system of d-dimensional pseudo-coordinates for every point and learn both the filters and patch operators that work on these coordinates, rather than using fixed kernels. Further, the authors of [5] propose edge-based convolutional kernels and dynamic graph updates. While being flexible and effective for general tasks these methods do not directly take the advantage of the knowledge of the projective geometry that we have in the specific context of the omnidirectional images considered in our work. Instead, we propose to model this a priori knowledge using a specifically designed graph representation. We further introduce a new way of creating such a representation and incorporating it inside a neural network for effective representation learning.
>
> A different method was suggested by [6], where the authors propose to process the directed graph by exploiting local graph motifs, which represent its connectivity patterns. The main differences of this method with our work are that first, this method assumes that the directed graph is already given. In our problem, building such a graph that is able to fully take advantage of the image projective geometry is actually one of the main contributions. Second, the approach in [6| does not use the knowledge of the coordinate system associated with omnidirectional images, which we however use in our architecture, in order to define filter orientations.
>
> Overall, they are truly key differences between the cited works and ours, which provides a constructive solution for the specific case of omnidirectional images. We will do our best to clarify it in the final version of the paper however.

---

### Meta-Review · Area_Chair1 · 2018-12-16
**Area chair recommendation**

**Confidence:** 5
**Recommendation:** Reject

**Metareview:**

Strengths:

This paper proposed to use graph-based deep learning methods to apply deep learning techniques to images coming from omnidirectional cameras.

Weaknesses:

The projected MNIST dataset looks very localized on the sphere and therefore does not seem to leverage that much of the global connectivity of the graph
All reviewers pointed out limitations in the experimental results.
There were significant concerns about the relation of the model to the existing literature.  It was pointed out that both the comparison to other methodology, and empirical comparisons were lacking.


The paper received three reject recommendations.  There was some discussion with reviewers, which emphasized open issues in the comparison to and references to existing literature as highlighted by contributed comment from Michael Bronstein.  Work is clearly not mature enough at this point for ICLR, insufficient comparisons / illustrations